# Failure of *Micractinium simplicissimum* Phosphate Resilience upon Abrupt Re-Feeding of Its Phosphorus-Starved Cultures

**DOI:** 10.3390/ijms24108484

**Published:** 2023-05-09

**Authors:** Elena Lobakova, Olga Gorelova, Irina Selyakh, Larisa Semenova, Pavel Scherbakov, Svetlana Vasilieva, Petr Zaytsev, Karina Shibzukhova, Olga Chivkunova, Olga Baulina, Alexei Solovchenko

**Affiliations:** 1Department of Bioengineering, Faculty of Biology, Lomonosov Moscow State University, 1-12 Leninskie Gory, 119234 Moscow, Russia; elena.lobakova@gmail.com (E.L.); ogo439@mail.ru (O.G.); i-savelyev@mail.ru (I.S.); semelar@mail.ru (L.S.); cyano@mail.ru (P.S.); vankat2009@mail.ru (S.V.); zaytsevpa@my.msu.ru (P.Z.); shibzukhova.karina@yandex.ru (K.S.); olga.chivkunova@mail.ru (O.C.); baulina@inbox.ru (O.B.); 2Institute of Natural Sciences, Derzhavin Tambov State University, Komsomolskaya Square 5, 392008 Tambov, Russia

**Keywords:** *Micractinium*, inorganic phosphate, polyphosphate, phosphorus toxicity

## Abstract

Microalgae are naturally adapted to the fluctuating availability of phosphorus (P) to opportunistically uptake large amounts of inorganic phosphate (P*_i_*) and safely store it in the cell as polyphosphate. Hence, many microalgal species are remarkably resilient to high concentrations of external P*_i_*. Here, we report on an exception from this pattern comprised by a failure of the high P*_i_*-resilience in strain *Micractinium simplicissimum* IPPAS C-2056 normally coping with very high P*_i_* concentrations. This phenomenon occurred after the abrupt re-supplementation of P*_i_* to the *M. simplicissimum* culture pre-starved of P. This was the case even if P*_i_* was re-supplemented in a concentration far below the level toxic to the P-sufficient culture. We hypothesize that this effect can be mediated by a rapid formation of the potentially toxic short-chain polyphosphate following the mass influx of P*_i_* into the P-starved cell. A possible reason for this is that the preceding P starvation impairs the capacity of the cell to convert the newly absorbed P*_i_* into a “safe” storage form of long-chain polyphosphate. We believe that the findings of this study can help to avoid sudden culture crashes, and they are also of potential significance for the development of algae-based technologies for the efficient bioremoval of P from P-rich waste streams.

## 1. Introduction

Phosphorus (P) is a major nutrient central to the processes of energy and information storage and exchange in cells [1,2,3]. Most of the habitats accessible to microalgae are characterized by the variable availability of P [4,5], but inorganic P species (referred to below as P*_i_*) are usually present at limiting concentrations [6,7,8]. To withstand prolonged P shortage, microalgae developed a set of adaptations collectively called “luxury uptake” [6,7]. These adaptations include the capability of absorbing P*_i_* in amounts much greater than their metabolic demand [9]. Another set of mechanisms is converting the newly acquired P*_i_* into relatively metabolically inert polyphosphate (PolyP) and storing it in the cell vacuole serve to avoid fatal displacement of the equilibria of vital metabolic reactions in which P*_i_* participates [1,3].

Possibly as a side effect of these adaptations, microalgae became remarkably resilient to high levels of external P*_i_* by far exceeding its environmental concentrations (1 g L^−1^ and more [10]). Some microalgal species found in P-polluted sites display very high P*_i_* tolerance (see, e.g., [10]). It makes microalgae a powerful vehicle for the biocapture of P from waste streams [11], thereby increasing the sustainability of using P resources, which are currently notoriously low (<20%) [12]. Indeed, there are numerous reports on the successful use of microalgae exerting luxury uptake of P [13,14] for the biotreatment of waste streams to avoid eutrophication [12,15] and produce environmentally friendly biofertilizers [16,17].

From a practical view, microalgae-based approaches for P biocapture from waste streams offer advantages over conventional techniques such as Enhanced Phosphorus Bioremoval, EPBR [2,18]. Using waste and side streams as P sources for industrial cultivation of microalgae makes bulk bioproducts such as P biofertilizers [16,19] or biofuels economically viable [20]. Biomass of microalgae is a potential source of PolyP that can be used in medicine for the development of biomaterials and in the food industry [21,22].

Despite the promise of microalgae-based P capture, its use is hindered by insufficient knowledge of luxury P uptake mechanisms [1,3] and the lack of strains resilient to high concentrations of P*_i_,* since many algal species commonly used in biotechnology can be already inhibited at a P*_i_* concentration above 0.1–0.3 g L^−1^ [23,24].

Nevertheless, there are reports on the toxicity of exogenic P*_i_* to microalgae [23,24,25], although the mechanisms of this phenomenon a far from being understood. Still, understanding P toxicity is important, e.g., for the development of viable biotechnologies for nutrient biocapture from P-rich waste streams and/or highly nutrient-polluted sites. To further bridge this gap, we investigated the failure of high P resilience in P-starved cultures.

We hypothesized that P toxicity can be linked to the formation of short-chain PolyP in the cell upon abrupt transition from P starvation to ample conditions. The *Micractinium simplicissimum* strain IPPAS C-2056 recently isolated from a P-polluted site served as a model organism for this study. The cultures of the *M. simplicissimum* grown in P-replete media exhibit a very high P*_i_* resilience [10]. In our recent experiments on P starvation, we observed a sudden culture death upon replenishment of P*_i_* to the P-depleted culture of the *M. simplicissimum*. Here, we report on the effect of an abrupt increase in external P*_i_* on the cell viability, P*_i_* uptake, and internal content of P and PolyP. We elaborate on possible mechanisms of P*_i_*-induced death of microalgal cells acclimated to P deficiency. Special attention was paid to the potential role of short-chain PolyP in these mechanisms.

## 2. Results

To test the effect of P starvation with subsequent P*_i_* replenishment on the *M. simplicissimum* culture, the P-sufficient culture was starved of P for 14 days, then P*_i_* was replenished to the medium, and the culture was monitored for another 10 days (see Section 4).

### 2.1. The Dynamics of the Culture Condition during Phosphorus Starvation and Re-Feeding

During 14 days of P starvation, the precultures gradually changed their color from green to yellow-green (Figure 1), which is typical of nutrient-starved cultures responding to the stress by a reduction in their photosynthetic apparatus. During the first week of P starvation, the culture exhibited a vigorous cell division commensurate to that commonly observed in P-replete cultures of the *M. simplicissimum*. The P-starved cultures demonstrated the onset of the stationary phase due to the P limitation by the 10th day of incubation.

Inorganic phosphate, P*_i_*, was replenished to the P-starved cultures to the final concentration of 0.8 g L^−1^, which is far below the level potentially toxic to the *M. simplicissimum* [10]. Since the pH of the cultures remained in the range of 6.7–7.7 throughout the experiment, the P*_i_* added to the cultures is expected to be readily available for uptake by the microalgal cells.

Our analysis of the absorbance spectra of the P-starved cultures (Figure 2) revealed an increase in the relative contribution of the absorption of pigments in the blue-green region of the visible part of the spectrum, which is consistent with the increase in the carotenoid-to-chlorophyll ratio (not shown) observed in this culture manifesting itself as the changes in coloration described above (see also Figure 1b).

The absorbance spectra recorded for the first 24 h after P*_i_* replenishment revealed a synchronous increase in the light absorption by the culture in the blue and red regions of the spectrum (Figure 2, curves 1 and 2) attributable to the accumulation of pigments and balanced culture growth. A total of 24 h after P*_i_* replenishment, the cultures started to show visual signs of damage such as discoloration and increased turbidity (see culture (iii) in Figure 1b) indicative of the presence of cell debris. The corresponding absorbance spectra revealed a profound bleaching of the photosynthetic pigment absorption bands (Figure 2, curves 3–5). The cultures became whitish by the 10th day of cultivation (Figure 1b and Figure 2). At the same time, the red absorption band of chlorophyll became undetectable on the spectra of the cultures, although the small peaks attributable to the absorption by carotenoids remained (Figure 2, curve 5). The appearance and the optical properties of the cultures almost did not change until the end of the observation period. Overall, these observations evidenced the progressive lysis of the microalgal cells upon replenishment of P*_i_* to the P-depleted cultures of the *M. simplicissimum,* which was confirmed by microscopic observations (see below).

### 2.2. Phosphorus Removal from the Medium and Its Uptake by the Cells

Monitoring of the removal of P*_i_* after its replenishment to the P-starved cultures showed that most (75–80%) of the added P*_i_* remained in the medium at all stages of the experiment (Figure 3). Around 3–5% of the added P*_i_* was adsorbed by the superficial structures of the *M. simplicissimum* cells as well as by cell debris (see below); the amount of the adsorbed P*_i_* was relatively constant.

The amount of P*_i_* taken up by the cells gradually increased over the first three days of the experiment, but it later started to decline with the corresponding increase in the external P*_i_* concentration (Figure 3). The increase in the residual P*_i_* in the culture likely reflected the release of P from the ruptured dead cells whose proportion increased in the culture, according to our observations outlined above.

The analysis of the total intracellular P content revealed a rapid increase in the dry weight (DW) percentage of P from 0.25% typical of P-starved cells to 4% DW during the first 24 h after replenishment of P*_i_* (Figure 4). The discrepancies between the trends shown in Figure 4 might stem from the difference in the techniques of sample processing (for more details, see Section 4). Another plausible reason is the shortage of metabolic resources of the cell available for the conversion of the incoming P*_i_* to PolyP. Later, the internal P content of the microalgal cells declined, on average to 1%, likely due to the predominant lysis of the cells, which took up the highest amounts of P*_i_*.

The PolyP content of the cells also increased rapidly during the first few hours after P*_i_* replenishment, but it reached its maximum of 0.35% DW within ca. four hours. This parameter returned to the initial level of ca. 0.02% DW by the end of the first day of incubation of the culture under P-replete conditions. The PolyP content of the cells did not change significantly thereafter but tended to decline (Figure 4).

### 2.3. Ultrastructural View of the Changes in the Cell P-Pools

To gain a deeper insight into the rearrangements of P pools in the culture of the *M. simplicissimum* under our experimental conditions, an electron microscopy study of cells sampled at the key stages of the experiment (see Figure 1) was carried out (Figure 5, Figure A1 and Figure A2).

The ultrastructure of the cells of *M. simplicissimum* constituting the preculture was similar to that of the cells grown under similar conditions [10]. Briefly, the preculture consisted of individual round-shaped cells containing a single nucleus and a parietal chloroplast with a centrally located pyrenoid; the culture also contained occasional autosporangia (Figure 5a and Figure 6a,b). Empty envelopes of the autosporangia and cell walls of dead cells were also encountered (Figure 6).

The surface of the cells and autosporangia lacked bristles. The cell wall did not reveal the characteristic trilamellar pattern (sporopollenin- or algenat-like layer). Cell walls of live and dead mature cells and the envelope of the autosporangia manifested, upon staining with DAPI, the yellow-green fluorescence indicative of the presence of PolyP (Figure 6b). However, this was not the case for the cell walls of the young cells.

The EDX spectra revealed the presence of P and iron in the cell walls as well as in the particles associated with them (Table A1). Carbon reserves were represented mainly by starch grains in the chloroplast (ca. 7.8 grains per cell section) and small (diameter 326 ± 18 nm, 986 nm max.) oleosomes (ca. 8 per cell section). Vacuoles contained spectacular inclusions—large round-shaped P-containing inclusions of non-uniform electron density (Figure 5a). Their EDX spectra possessed a distinct peak of P along with the characteristic peaks of nitrogen, calcium, and magnesium (Table A1, Figure 7a). The vacuoles also contained small roundish, electron-opaque inclusions harboring P or P in combination with uranium (see below and Table A1). Moreover, small electron-opaque spherules (10–40 nm in diameter; 5–20 instances per cell section) of the same composition were detected in the cytoplasm (Figure A1a, Table A1). Roundish inclusions were in the cells and in the sporangia. Roundish inclusions located in the cells and in the sporangia revealed, upon staining with DAPI, the yellow-green fluorescence characteristic of PolyP (Figure 6a,b).

The starvation of P profoundly influenced the cell morphology and ultrastructure (Figure 5b). The amount of cells retaining their structural integrity declined by 10–15% (76% vs. 92% in the preculture). The thylakoids remaining in the granae were moderately expanded (the lumen width was up to 20 nm vs. 5–7 nm in the preculture; Figure A1a,b). The total number of starch grains decreased by 27%. At the same time, the number of oleosomes increased by 35%. Moderately electron-dense oleosomes were located on the periphery of the cell and merging, forming large oleosomes (ca. 3350 nm in diameter; Figure 5b). These rearrangements were in accord with the observed decline in chlorophyll (Figure 2a).

The vacuolar inclusions also displayed considerable changes during P starvation of the culture. They were represented by large amoeboid globules with moderately increased electron density (Figure 5b and Figure A1c) and sickle-shaped elongated or reticular zones of high electron density or aggregations of small particles (Figure 5b) as well as by fragments of structures resembling a multiwire cable (Figure A1d). Small electron-dense P-containing globules were occasionally found in the vacuoles of the P-starved cells (Table A1).

Interestingly, the DAPI-stained P-starved cells, which ceased to divide, envelopes of the sporangia, and the cell debris retained the characteristic yellow-green fluorescence (Figure 6c,d). The EDX spectra of the structures (cell wall, different types of vacuolar inclusions, and small cytoplasmic electron-opaque spherules in the stroma of chloroplast) revealed the peaks of P. The spectra were also accompanied by the peaks of other elements typically found together with PolyP (calcium, magnesium, and sodium). These spectra also featured the peaks of sulfur or uranium stemming from the binding of the uranyl acetate (see Methods) with phosphate and carboxil groups of proteins and nucleic acids (Table A1, Figure 7). The magnitude of the P peak was lower than that of the N peak in the EDX spectra of the vacuolar large globules.

The analysis of ultrastructural traits of the cells after replenishment of P*_i_* to the culture revealed diverse signs of P uptake by the cells. During the first 24 h after P*_i_* re-feeding, the proportion of the cells retaining their structural integrity increased to 88–86%, the formation of autosporangia and autospore release resumed, and young cells appeared (Figure 5c–e and Figure 6).

The cell walls of the young cells were not stained with DAPI; accordingly, the P peak in their EDX spectra was low or absent. The cells in the cultures re-fed with P*_i_* were highly heterogeneous regarding their vacuolar inclusion condition. Most of these inclusions assumed a porous, sponge-like structure (Figure 5c,d and Figure A1e–g). This process was accompanied by a decline in the magnitude of N peaks and an increase in the magnitude of P peaks in the EDX spectra of these structures. In addition to this, electron-opaque globules and layers appeared inside the vacuole and on the inner surface of the tonoplast. The cells possessing these features also retained structurally intact chloroplasts (Figure 5c–e and Figure A1e–g). A sharp increase in the number (from several dozens to several hundred instances per cell section) of small (14–36 nm in diameter) spherules was also observed in the cytoplasm, mitochondria, nucleus, and dictyosomes of the Golgi apparatus. The EDX spectra of the inclusions, both vacuolar and extra-vacuolar, featured spectral details typical of PolyP including the peaks of P, magnesium, and calcium (Appendix A, Figure 7). The presence of abundant PolyP was also confirmed with DAPI staining (Figure 6).

The rest of the cell population, including autospores, displayed a gross accumulation of small granules in their cytoplasm; this phenomenon was accompanied by the degradation of all organelles (mitochondria, nucleus, vacuoles with their content, chloroplasts, etc.) (Figure 5c–e, Figure A1e–g and Figure A2, Appendix A). The amount of the destructed cells and cell debris increased with time after the P*_i_* re-feeding from 53% on the 3rd day to 77% on the 7th day and to 92% by the 10th day. The small electron-dense spherules remained at the location of the degraded organelles and cell walls; later, their number decreased from several hundreds to several dozens of instances. Surprisingly, the degraded cells retained abundant starch grains and oleosomes.

### 2.4. Responses of the Photosynthetic Apparatus of the Cells to P Starvation and Re-Feeding

The physiological condition of the microalgal cells as manifested by the functioning of their photosynthetic apparatus (PSA) during the different phases of the experiment was assessed by recording and analyzing the induction curves of chlorophyll *a* fluorescence. This analysis was based on the relevant parameters of the JIP test (Appendix A; Figure 8).

Acclimation of the culture to P starvation was accompanied by a small decline in the potential maximal photochemical quantum yield of photosystem II, Fv/Fm. The photosynthetic performance index, PI_ABS_ was more responsive as an indicator of the P starvation stress in *M. simplicissimum,* declining from ca. 0.4 to nearly 0. The flux of energy thermally dissipated by the PSA of the microalgal cells indicative of the engagement of photoprotective mechanisms, DI_0_/RC, increased very slightly. Likely, this was due to the concomitant decline in the number of reaction centers manifested by the decline in chlorophyll content (not shown; see also Figure 2). At the same time, the parameter reflecting Stern–Volmer quenching of chlorophyll fluorescence, NPQ, increased dramatically from 0 (typical of unstressed cells of P-sufficient preculture) to 1.5.

Collectively, the data on the condition of PSA (Figure 8, negative time values) suggested that the *M. simplicissimum* culture was apparently unaffected by the lack of available P in the medium for ca. 10 days (likely due to large intracellular P reserves). Later, the cells had rapidly (over three days) acclimated to the stress, mostly by adjusting their chlorophyll content and thermal dissipation of the absorbed light energy.

Replenishment of P*_i_* to the P-depleted *M. simplicissimum* culture triggered rapid, dramatic changes in the condition of the PSA of its cells (Figure 8, positive time values). Thus, Fv/Fm declined to the level of 0.1–0.2, and near-zero PI_ABS_ evidenced a near-complete lack of photosynthetic activity. The NPQ parameter declined rapidly and did not increase significantly thereafter. By contrast, a tremendous increase in DI_0_/RC and its variation was detected (again, likely to a gross decline in the number of reaction centers manifested by the decline in photosynthetic pigment content).

## 3. Discussion

To the best of our knowledge, this is the first report on a failure of tolerance to a high external concentration of P*_i_* in a microalga *Micractinium simplicissimum* IPPAS C-2056, which was previously shown to be highly tolerant to this stressor [10]. We attempted to link the actual level of tolerance to acclimation of the microalga to different levels of P in the medium. We also tried to infer a plausible hypothesis explaining these apparently controversial phenomena from the physiological and ultrastructural evidence collected during this study and backed up by the current knowledge of luxury P uptake and the physiological role of PolyP in microalgal cells.

Importantly, the phenomenon of failed P*_i_* tolerance was observed only when the P-starved *M. simplicissimum* culture was abruptly re-fed with P*_i_*. This phenomenon was observed despite the fact that the concentration of the P*_i_* added was far below the level potentially toxic to P-sufficient cultures of this microalga as was revealed by our earlier studies [10].

The dramatic response of *M. simplicissimum* to abrupt re-feeding with P*_i_* was accompanied by a peculiar pattern of changes in the distribution of P in the cells. There were other physiological manifestations (e.g., the changes in the photosynthetic apparatus) indicative of the acclimation state of the microalga. Overall, the acclimation of *M. simplicissimum* to P shortage at the first phase of the experiment manifested as the onset of mild stress as was documented in other microalgal species such as *Chlorella vulgaris* [26] and *Lobosphaera incisa* [27,28].

Specifically, a moderate reduction in the photosynthetic apparatus was observed as reflected by a decline in chlorophyll (Figure 2) and the accumulation of carbon-rich reserve compounds (Figure 5, Figure A1 and Figure A2) along with a depletion of P reserves in the cell. These rearrangements were accompanied by the up-regulation of photoprotective mechanisms based on thermal dissipation of the observed light energy, which is also typical of the acclimation of microalgae to nutrient shortage [26,28,29]. Nevertheless, the cells retained their structural integrity, and their photosynthetic apparatus remained functional despite a clearly observed expansion of the thylakoid lumens. Interestingly, the cells of the P-starved culture, which already ceased to divide, possessed a sizeable amount of PolyP granules and N-containing matter accumulated in their vacuoles similarly to that documented in other microalgae [27,30,31]. These P reserves are obviously represented by a slowly mobilizable fraction of PolyP, which frequently remains even in the P-starved cells [7,32]. Taken together, these observations suggest that the microalgal cultures experience only mild stress as a result of P starvation under our experimental conditions.

After re-feeding of the P-starved culture of *M. simplicissimum* with 800 mg L^−1^ P*_i_*, up to 20% of the added P*_i_* was gradually removed from the medium by the cells by the 3rd day of incubation (Figure 3). Approximately 5% of the added P*_i_* was reversibly adsorbed on the surface of the cells. DAPI staining revealed a characteristic yellow fluorescence localized in the cell wall (Figure 6). This observation is in line with the previously documented ability of this strain to adsorb P*_i_* and form P-containing nanoparticles on its surface structures [10]. It corroborates previous reports on the dynamics of P depots in cell walls of diverse organisms including fungi, bacteria [21,33], and microalgae [34]. As revealed by EDX spectroscopy, PolyP is characterized by co-localization with calcium, magnesium, or (less frequently) potassium and sodium [35,36,37]. In certain cases, we also observed the P peak in combination with that of uranium. Since the uranyl cation used for the sample preparation binds to phosphate and carboxyl groups [38], this can be a manifestation of phosphorylated proteins and nucleic acids in these cell compartments.

The amount of P internalized by the cells as well as the amount of intracellular PolyP also increased during the first four hours after re-feeding. Later, the amount of total intracellular P remained at the level of 4% of cell dry weight, but the PolyP content started to decline. (Notably, the method of PolyP assay used here is optimized for long-chain PolyP, and the internal PolyP content can be slightly underestimated since a part of short-chain PolyP can escape detection.) At the same time, the EDX spectral signature of PolyP was still detected (Figure 7 and Table A1).

Starting from the 1st day of incubation, the progressive signs of cell rupture were recorded. Thus, a recovery of the photosynthetic apparatus would be expected normally after P*_i_* replenishment as was normally the case [1,2,3]. Instead of this, we observed a complete failure of photoprotective mechanisms, indicative of acute damage to the cell similar to the effect of severe stress or a toxicant at a sublethal concentration [39]. Confronting the observed effects with the reports on P*_i_* toxicity found in the literature [23,40], we hypothesized that short-chain PolyP might be involved in the massive cell death observed in *M. simplicissimum* after its re-feeding with P*_i_* following P starvation.

A possible scenario of the short-chain PolyP-mediated P*_i_* toxicity might involve the following steps. First, the P-starved microalgal cells deploy, as a common pattern of nutrient shortage acclimation, the mechanisms making them capable of fast P*_i_* uptake [1,6,7,26]. At the same time, their capability of photosynthesis becomes impaired because of the reduction in the photosynthetic apparatus (see also Figure 8). Upon re-feeding of the culture acclimated to P shortage, a large amount of P*_i_* rapidly enters the cell. As a result, the biosynthesis of PolyP is triggered, since PolyP serves as a buffer for the storage of P*_i_* when it becomes available [41,42]. However, the cell acclimated to a nutrient shortage is, to a considerable extent, metabolically quiescent (in particular, its photosynthetic apparatus is downregulated, and a large part of the light energy it captures is dissipated into heat). At the same time, the biosynthesis of PolyP is very energy-intensive, and this energy comes mostly from photosynthesis [43]. As the net result, these processes trigger the mass accumulation of short-chain PolyP, but the newly formed PolyP cannot be further elongated due to the metabolic restrictions mentioned above. Overall, the toxic effect of the short-chain PolyP rapidly accumulated in all compartments of the cell leads to its damage and, eventually, death, which was the case under our experimental conditions.

It should be noted in addition that the barrier function of the cell wall regarding P*_i_* uptake is an important determinant of the P*_i_* resilience in *M. simplicissimum* [10]. P*_i_* re-feeding of the culture that was pre-starved of P triggers cell division. Therefore, the proportion of young cells in the cell population increases, whose cell wall can be less efficient as a barrier to P*_i_* uptake than the cell wall of mature cells. This can render the young cells more vulnerable to the surge of P*_i_* into the cell.

The hypothesis outlined above can explain the observed phenomenon of the failed P*_i_* tolerance by analogy with the toxic effect of short-chain PolyP initially described in yeast cells [40], which was also implied in *Chlorella regularis* [23]. In these works, disorganization of the cell structure has been proposed as a major hallmark of elevated P*_i_* toxicity mediated by PolyP. This hypothesis is also supported by the presence of genes encoding the PolyP polymerases from the VTC family [1,2,3] potentially involved in the synthesis of the short-chain PolyP [39] in the genome of a closely related representative of the genus *Micractinium*, *M. conductrix* [44]. A homologue of one of these genes was putatively discovered in our pilot studies of the species used in this work, *M. simplicissimum* (in preparation).

It was also demonstrated recently that PolyP that accumulates outside the cell vacuole is the main factor of PolyP-mediated toxicity of elevated external P*_i_* [25]. One can speculate that, mechanistically, the short-chain PolyP, when formed outside the vacuole, can interfere with protein folding and/or the matrix synthesis of biomolecules. This capability of interacting with important polymeric biomolecules is a typical trait of PolyP as the “molecular fossil” retained from ancient times when they were potentially involved in the primordial matrix synthesis and the genesis of life [45].

Finally, we would like to underline the importance of understanding the limits of the high tolerance of microalgae to elevated levels of external P*_i_* not only for basic science but also from the practical standpoint as well. One should consider that abrupt changes in P availability can cause the normally high P*_i_* tolerance of microalgae to fail and lead to a sudden culture crash. This is possible, particularly in wastewater treatment facilities during the injection of a new portion of P-rich wastewater into a P-depleted culture. Nevertheless, the toxic effects of P*_i_* in microalgae remain quite underexplored. Further research in this direction might include studies of the genetic control and implementation of the formation of PolyP of variable chain length in different microalgal species as a function of P*_i_* availability.

## 4. Materials and Methods

### 4.1. Strain and Cultivation Conditions

Unialgal culture of an original chlorophyte *M. simplicissimum* strain IPPAS C-2056 served as the object of this study. The preculture was grown in 750 mL Erlenmeyer flasks with 300 mL of modified BG-11 [10,46] medium designated below as BG-11_K_ medium. The microalgae were cultured at 25 °C, and continuous illumination of 40 μmol m^−2^ s^−1^ PAR quanta was provided by daylight fluorescent tubes (Philips TL-D 36W/54-765). The cultures were mixed manually once a day.

To obtain P-depleted cultures of *M. simplicissimum*, the preculture cells were harvested by centrifugation (1000× *g*, 5 min), twice washed by the BG-11_K_ medium lacking P and resuspended in the same medium to the chlorophyll concentration and biomass content of 25 mg·L^−1^ and 0.4 g·L^−1^, respectively, in 0.6 L glass columns (4 cm internal diameter) containing 400 mL of the cell suspension. The columns were incubated in a temperature-controlled water bath at 27 °C with constant bubbling with a 5% CO_2_: 95% air mixture prepared and delivered at a rate of 300 mL·min^−1^ (STP). Air passed through a 0.22 μm bacterial filter (Merck-Millipore, Billerica, MA, USA) and pure (99.999%) CO_2_ from cylinders were used. A continuous illumination of 240 μmol PAR photons · m^−2^ · s^−1^ by a white-light-emitting diode source as measured with a LiCor 850 quantum sensor (LiCor, Lincoln, NE, USA) in the center of an empty column was used. Culture pH was measured with a bench-top pH meter pH410 with a combined electrode ESK-10601/7 (Aquilon, St.-Petersburg, Russia).

The cultures were considered to be P-depleted (having the minimum cell P quota) when the culture biomass did not increase consecutively for three days and showed a decline in chlorophyll content (for details on the monitoring of the corresponding parameters, see below). To ensure that the lack of P was the only limiting factor, the P-starved culture was diluted with BG-11_K_ medium to keep their OD_678_ below 1.5 units, and the residual nitrate content was checked periodically (see below). P*_i_* was replenished to the stationary P-depleted cultures in the form of KH_2_PO_4_ to a final P*_i_* concentration of 0.8 g L^−1^, and the cultures were monitored (see below) for another 10 days.

### 4.2. Suspension Absorption Spectra and Pigment Assay

Absorbance spectra of the microalgal cell suspension samples were recorded using an Agilent Cary 300 UV-Vis spectrophotometer (Agilent, Santa Clara, CA, USA) equipped with a 100 mm DR30A integrating sphere of the same manufacturer and corrected for the contribution of light scattering [47]. Pigments from the cells were extracted by dimethyl sulfoxide (DMSO) and quantified spectrophotometrically using the same spectrophotometer. The content of chlorophyll *a* and *b* and total carotenoids was calculated using the equations from [48].

### 4.3. Biomass Content Determination

Dry cell weight (DCW) was determined gravimetrically. Routinely, the cells deposited on nitrocellulose filters (24 mm in diameter and 0.22 μm pore size) Millipore (Merck-Millipore, Billerica, MA, USA) were oven-dried at 105 °C to a constant weight and weighed on a 1801MP8 balance (Sartorius GmbH, Gottingen, West Germany). In certain experiments, the 1.5 mL aliquots of the culture were gently centrifuged (1000× *g*, 5 min) in pre-weighed microtubes, the supernatant was discarded, and the pellet was dried under the same conditions. The tubes with the dried cell pellet were closed and weighed on the same balance.

### 4.4. Assay of External and Intracellular Content of Different P Species

The concentration of P*_i_*, along with that of nitrate, was determined in a cultivation broth, and cell wash liquid was assayed essentially as described in our previous work [10] using a Dionex ICS1600 (Thermo-Fisher, Sunnyvale, CA, USA) chromatograph with a conductivity detector, an IonPac AS12A (5 μm; 2 × 250 mm) anionic analytical column, and an AG12A (5 μm; 2 × 50 mm) guard column according to the previously published protocol [49].

To determine the P*_i_* adsorption of the cultures, the cells were twice washed with a 15 mL BG-11_K_ medium lacking phosphorus. According to our previous data on independent ^32^P-NMR control on the completeness of the P*_i_* wash-off [26], it was sufficient to remove > 99% of the adsorbed P*_i_*. The wash liquid batches obtained from washing the same sample were pooled. The P*_i_* content of the pooled wash liquid was assayed with HPLC as described above and confirmed independently with the molybdenum blue method [50].

A modified method by Ota and Kawano [49] was used for the total intracellular P and PolyP determination. Briefly, the cell pellets from 15 mL aliquots of MA suspension after washing and removing absorbed P were resuspended in 3 mL of distilled water. The suspended samples were divided and transferred to 2 mL microtubes: 2 mL for the PolyP assay and 1 mL for the total-P assay. The PolyP was extracted from cells with 5% sodium hypochlorite and precipitated with ethanol [51]. A total of 500 µL of distilled water and 100 µL of 4% (*w*/*v*) potassium persulfate were added to the precipitated PolyP. For hydrolysis to orthophosphate, the PolyP solution was autoclaved at 121 °C for 20 min. For the total P assay, the cell pellets were disrupted in 1 mL of distilled water with G8772 glass beads (Sigma-Aldrich, St. Louis, MO, USA) and 200 µL of 4% (*w*/*v*) potassium persulfate added by vigorous mixing on a V1 vortex (Biosan, Riga, Latvia) for 15 min at 4 °C and subsequent autoclaving at 121 °C for 20 min. After centrifugation of the autoclaved samples (3000× *g*, 5 min), the P*_i_* concentration was assayed in the supernatants using the molybdenum blue method [50].

### 4.5. Light Microscopy

Light microscopy was carried out using a Leica DM2500 microscope equipped with a digital camera Leica DFC 7000T and the light filter set AT350/50xT400lp ET500/100m (Leica, Wetzlar, Germany). PolyP inclusions were visualized using vital staining with fluorescent dye 4′,6-diamidino-2-phenylindole (DAPI) dissolved in dimethyl sulfoxide [28]. The cells were incubated in a 0.05% water solution (*w*/*v*) of the dye for 5−10 min at room temperature, washed with water, and studied with the microscope in brightfield mode.

### 4.6. Transmission Electron Microscopy

The microalgal samples for transmission electron microscopy (TEM) were prepared according to the standard protocol as described previously [52]. The cells were fixed in 2% *v*/*v* glutaraldehyde solution in 0.1 M sodium cacodylate buffer (pH 7.2–7.4, depending on the culture pH) and then post-fixed for 4 h in 1% (*w*/*v*) OsO_4_ in the same buffer. The samples, after dehydration through graded ethanol series including anhydrous ethanol saturated with uranyl acetate, were embedded in araldite. Ultrathin sections were made with Leica EM UC7 ultratome (Leica Microsystems, Wetzlar, Germany), mounted to the formvar-coated TEM grids, stained with lead citrate according to Reynolds [53], and examined under JEM-1011 and JEM-1400 (JEOL, Tokyo, Japan) microscopes. All quantitative morphometric analyses were performed as described previously [52]. Briefly, at least two samples from each treatment were examined on cell sections made through the cell equator or sub-equator. The subcellular structures and inclusions were counted on the sections. Linear sizes of the subcellular structure were measured on the TEM micrographs of the cell ultrathin sections (*n* ≥ 20) using Fiji (ImageJ) v. 20200708-1553 software (NIH, Bethesda, MA, USA).

### 4.7. Analytical Electron Microscopy

The samples for nanoscale elemental analysis in analytical TEM using energy-dispersive X-ray spectroscopy (EDX) were prepared as described previously [29]: fixed, dehydrated, and embedded in araldite as described above except the staining with lead citrate. Semi-thin sections were made with Leica EM UC7 ultratome (Leica Microsystems, Wien, Germany) and examined under a JEM-2100 (JEOL, Japan) microscope equipped with a LaB_6_ gun at the accelerating voltage of 200 kV. Point EDX spectra were recorded using a JEOL bright-field scanning TEM (STEM) module and X-Max X-ray detector system with an ultrathin window capable of analysis of light elements starting from boron (Oxford Instruments, Abingdon, UK). The energy range of recorded spectra was 0–10 keV with a resolution of 10 eV per channel. At least 10 cells per specimen were analyzed. Spectra were recorded from different parts of electron-dense inclusions and from other (sub)compartments of microalgae cells. Spectra were processed with INKA software (Oxford instruments, Abingdon, UK) and presented in the range of 0.1–4 keV.

### 4.8. Photosynthetic Activity and Photoprotective Mechanism Assessment

Estimations of the photosynthetic activity of the microalgal cells dark-adapted for 15 min were obtained by recording Chl *a* fluorescence induction curves by using an FP100s portable PAM fluorometer (PSI, Czech Republic) using the built-in protocol supplied by the manufacturer. The recorded curves were processed by the built-in software of the fluorometer, and the JIP test parameters indicative of the functional condition of the photosynthetic apparatus of the microalgal cells were calculated (Appendix A, see also [54,55]).

### 4.9. Statistical Treatment

Under the specified conditions, three independent experiments were carried out for each treatment repeated in duplicate columns. The average values (*n* = 6) and corresponding standard deviation are shown unless stated otherwise.

## 5. Conclusions

Microalgae including the *M. simplicicssimum* strain studied here are resilient to very high concentrations of exogenic P*_i_*. As it turned out in this work, this resilience fails after the abrupt re-supplementation of P*_i_* to the culture pre-starved of P. This was the case even if P*_i_* was re-supplemented at a concentration far below the level that can be toxic to the P-sufficient culture. The obtained evidence suggests that this effect can be mediated by the rapid formation of the potentially toxic short-chain PolyP following the mass influx of P*_i_* into the P-starved cell. A possible reason for this is that the preceding P starvation impairs the capacity of the cell to convert the newly absorbed P*_i_* into a “safe” storage form of long-chain PolyP. We believe that the findings of this study can help to avoid sudden culture crashes and are of potential significance for the development of algae-based technologies for efficient bioremoval of P from P-rich waste streams.

## Figures and Tables

**Figure 1 ijms-24-08484-f001:**
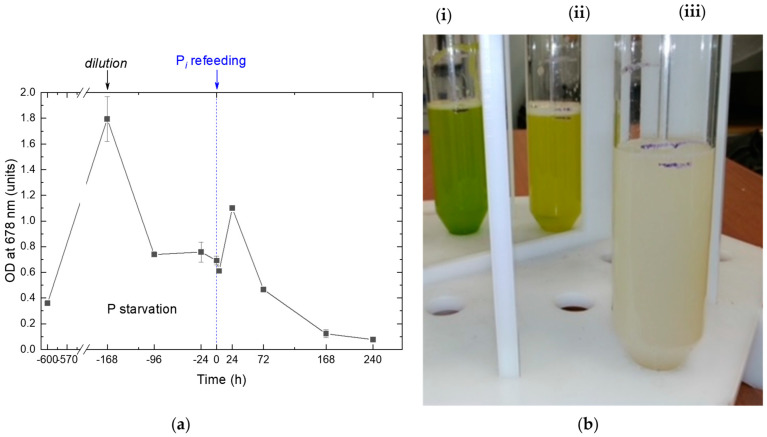
The changes in the *Micractinium simplcissimum* IPPAS C-2056 culture condition as manifested by (**a**) the kinetics of changes in the OD_678_ during its phosphorus starvation (negative time values) and after re-feeding of the P-starved cultures with P*_i_* (positive time values). The moment of P*_i_* re-feeding (*t* = 0 h) is marked on the graph by a vertical dashed line. (**b**) A typical visual appearance (left to right) of *Micractinium simplcissimum* IPPAS C-2056 (**i**) preculture; (**ii**) culture P-starved for 14 days; and (**iii**) the culture 168 h after re-feeding with P*_i_* (see also Figure 2). Cultures from different time-shifted replicas of the experiment running in parallel are shown in (**b**).

**Figure 2 ijms-24-08484-f002:**
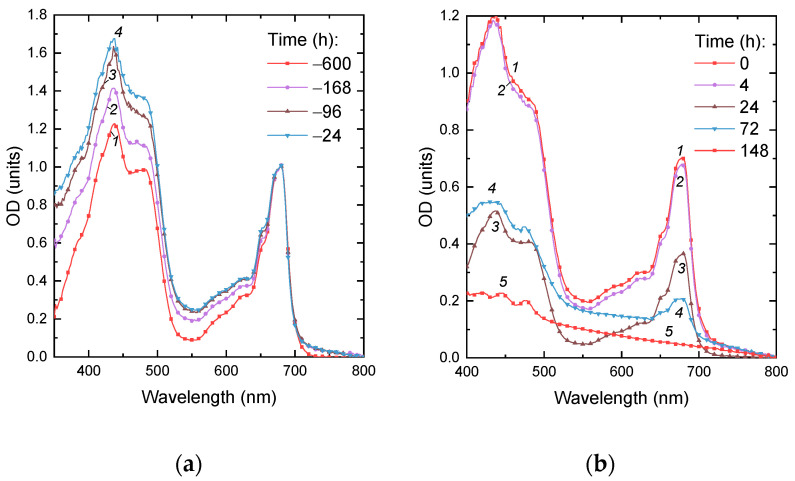
Changes in absorption spectra of the *Micractinium simplcissimum* IPPAS C-2056 cultures during (**a**) its P starvation and (**b**) after re-feeding of the pre-starved *Micractinium simplcissimum* IPPAS C-2056 cultures with P*_i_*. The time of P starvation indicated in the panels (h) is counted down to the moment of P*_i_* re-supplementation to the culture (*t* = 0 h; see Figure 1).

**Figure 3 ijms-24-08484-f003:**
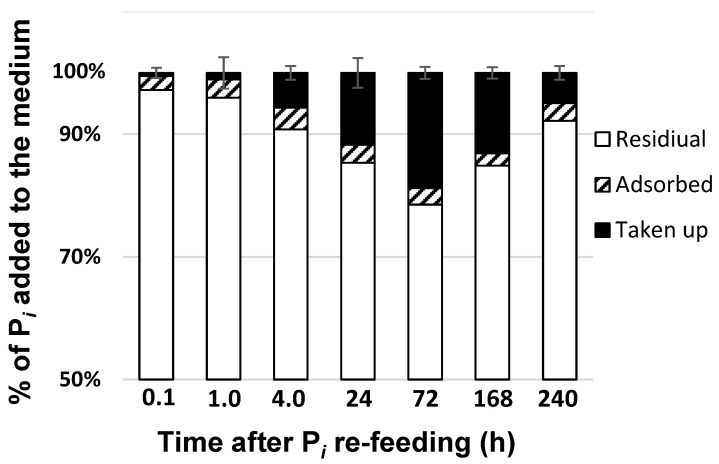
Changes in the distribution of P*_i_,* which was either taken up or adsorbed by the cells of *Micractinium simplicissimum* IPPAS C-2056 or remained in the medium after re-feeding of P*_i_* to the P-starved cultures (see Figure 1).

**Figure 4 ijms-24-08484-f004:**
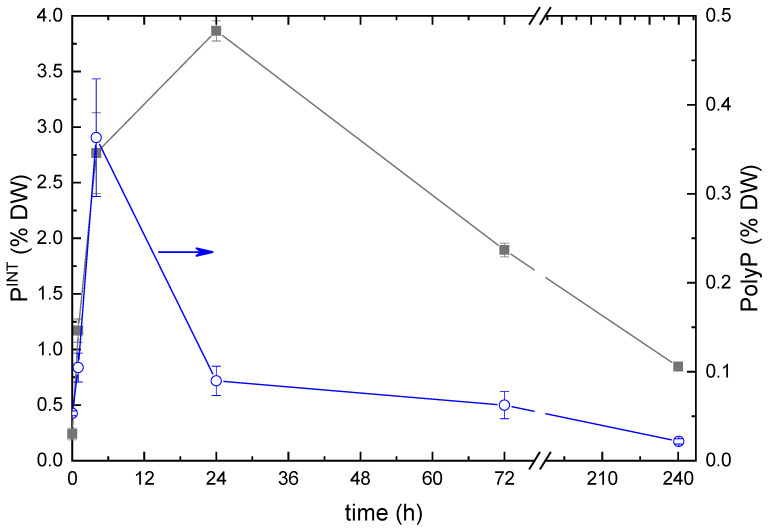
The changes in total intracellular P (**left** scale; black line) and PolyP (**right** scale; blue line) contents in the cultures of *Micractinium simplicissimum* IPPAS C-2056 after re-feeding of the P-starved cultures with P*_i_*.

**Figure 5 ijms-24-08484-f005:**
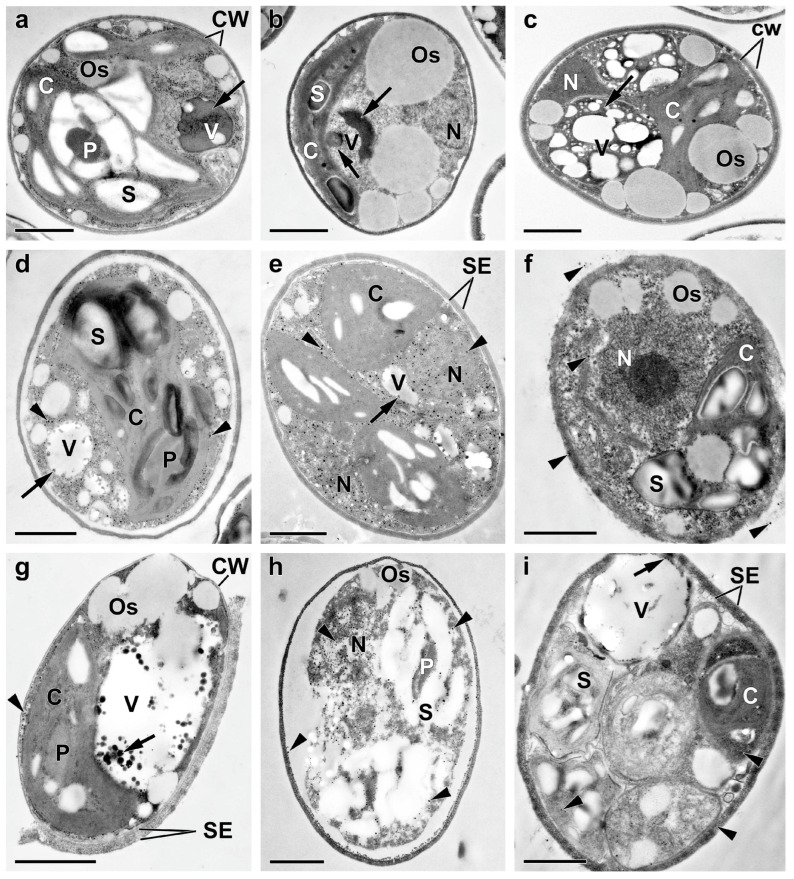
Typical transmission electron micrographs of cells (**a**–**d**,**f**–**h**) and autosporangia (**e**,**i**) of *Micractinium simplicissimum* IPPAS C-2056 reflecting their condition at different phases of the experiment (see also Figure 1). The micrographs of the culture in the BG-11_K_ medium in glass columns, which was incubated with constant bubbling with 5% CO_2_ (**a**), P-starved cells (**b**), and the cells (**a**–**d**,**f**–**h**) and sporangia (**e**,**i**) sampled 4 h (**c**), 24 h (**d**,**e**), and 72 h (**f**–**i**) after re-feeding of the P-starved cultures with P*_i_* are shown. C, chloroplast; CW, cell wall; N, nucleus; Os, oleosome; P, pyrenoid; S, Starch granule; SE, sporangium envelope; V, vacuole. The arrows point to the inclusions in the vacuole. The arrowheads point to the electron-opaque particles on/in the cell wall and them adsorbed on the cell surface of the clusters and also to the spherules in the cytoplasm, the nucleus, and destructed chloroplast. Scale bars = 1 μm.

**Figure 6 ijms-24-08484-f006:**
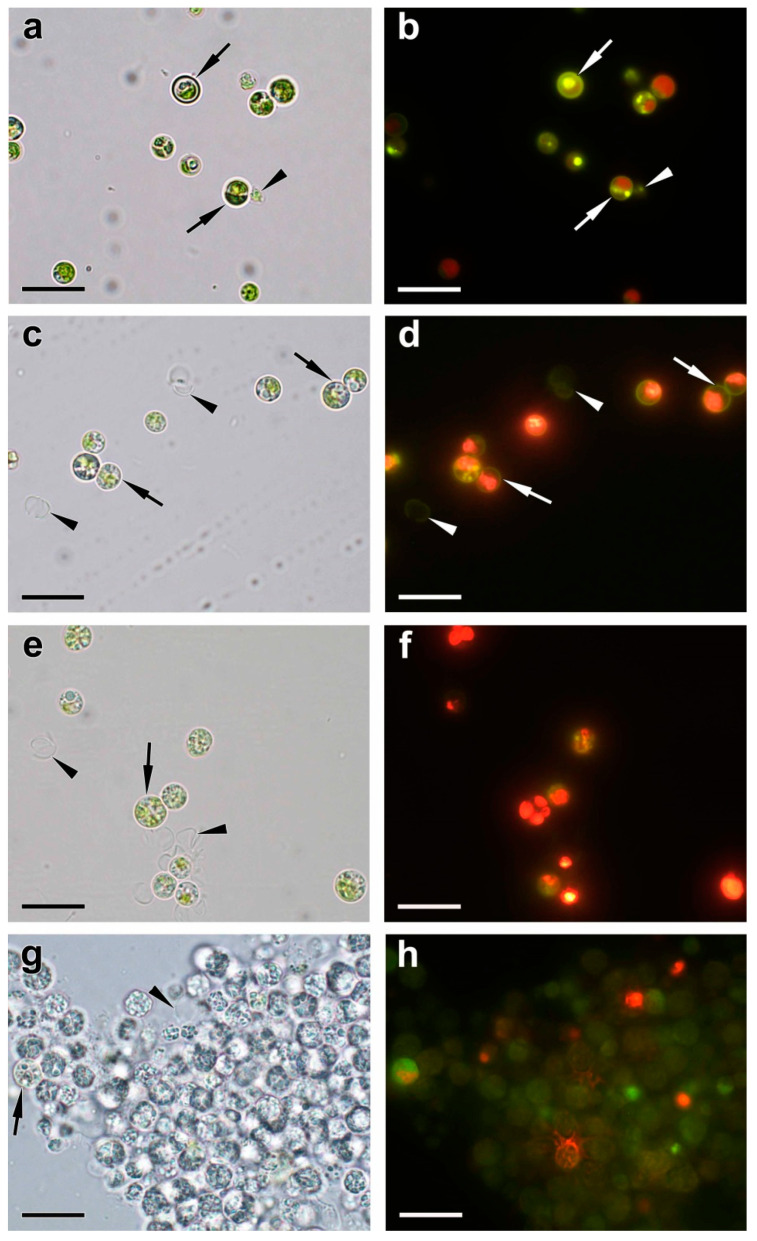
Typical brightfield (**a**,**c**,**e**,**g**) and fluorescent (**b**,**d**,**f**,**h**) microphotographs of DAPI-stained cells of *Micractinium simplicissimum* IPPAS C-2056 cells in preculture (**a**,**b**) and obtained 1 h (**c**,**d**), 24 h (**e**,**f**), and 168 h (**g**,**h**) after re-feeding of the P-starved culture with P*_i_*. The arrows point to the cell wall. The arrowheads point to the cell debris including walls of the destructed cells. Scale bars = 20 μm.

**Figure 7 ijms-24-08484-f007:**
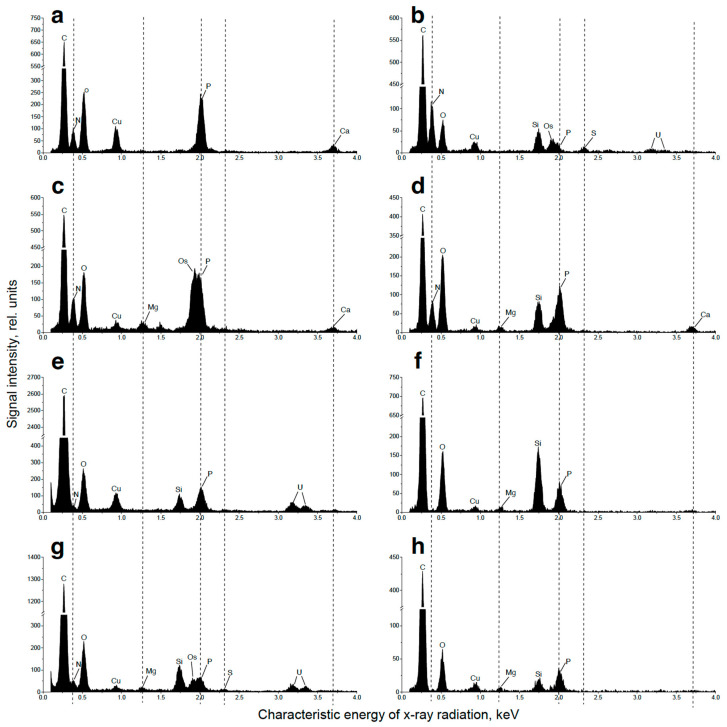
Representative EDX spectra of *Micractinium simplicissimum* IPPAS C-2056 cell structures potentially related to metabolism and storage of phosphorus at different phases of the experiment (see Figure 1). Vacuolar large globules in cell of the culture in the BG-11_K_ medium (**a**), in P-starved culture (**b**), and in the cells sampled 4 h after re-feeding of the P-starved cultures with P*_i_* (**c**); all show inclusions in cytosol (**d**), in the chloroplast stroma (**e**), the vacuolar small spherules (**f**), in the nucleus (**g**), and in the cell wall (**h**) after re-feeding of the P-starved cultures with P*_i_*. All the EDX spectra possessed characteristic peaks attributable to carbon (K_α_ = 0.28 keV) and oxygen (K_α_ = 0.53 keV), the major organic constituents of biological samples and the epoxy resin they were embedded in. The spectra also contained peaks of copper (L_α_ = 0.93 keV) from the copper grids used for the sample mounting, as well the peaks of osmium (M_β_ = 1.91 keV) and uranium (M_α_ = 3.16 keV, M_β_ = 3.34 keV) used for cell fixation. The peaks of silicon (K_α_ = 1.74 keV) and aluminum (K_α_ = 1.49 keV) originate from the microscope hardware background elements.

**Figure 8 ijms-24-08484-f008:**
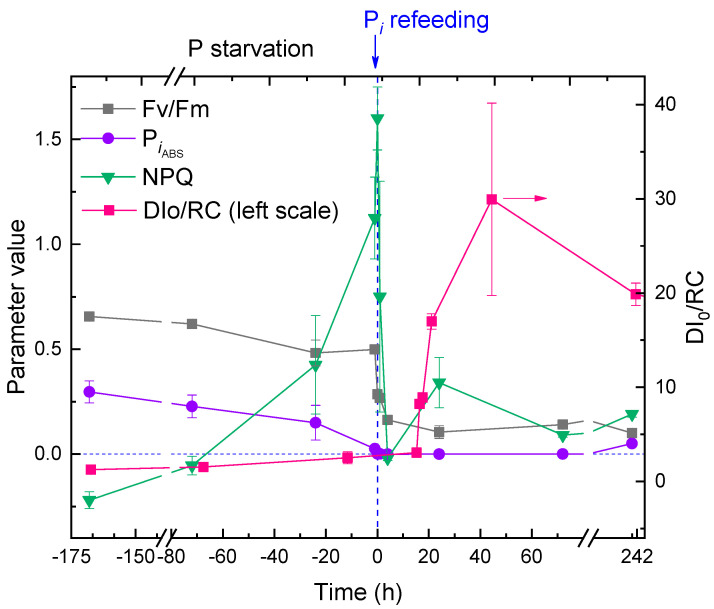
The kinetics of changes in JIP test (see Appendix A) parameters (potential maximal photochemical quantum yield of photosystem II, Fv/Fm; performance index, Pi_ABS_; the flux of thermally dissipated energy flux per reaction center, DI_0_/RC; left scale) and Stern–Volmer non-photochemical quenching (NPQ) in the cultures of *Micractinium simplcissimum* IPPAS C-2056 during its phosphorus starvation (negative time values) and after re-feeding of the P-starved cultures with P*_i_* (positive time values). The moment of P*_i_* re-feeding (*t* = 0 h) is specified on the graph.

## Data Availability

The data are available from the corresponding author upon a reasonable request.

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
