# Peer review of "Failure of Micractinium simplicissimum Phosphate Resilience upon Abrupt Re-Feeding of Its Phosphorus-Starved Cultures"

_ijms, 2023, doi:10.3390/ijms24108484_

Round 1
Reviewer 1 Report
“Failure of Micractinium simplicissimum phosphate resilience upon abrupt refeeding of its phosphorus-starved cultures” is quite promising. However, english is not readable. To strengthen the English expression, you should go with a native speaking english or adopt an english editing services.
Introduction should be based on your research with reference.
The hypothesis is not clear.
.
Result should be clearly described as per the experimental data obtained.
I don’t find any future direction of this work.
Discussion poorly written. Improve it.
Overall, this research is quite promising. Although the author did the convincing work, but needs to clarify. Considering all the things, this form of manuscript cannot be published prior to revision. Therefore, I recommended it for major revision.
Author Response
Introduction should be based on your research with reference.
RESPONSE: we believe it is the case for our manuscript (refs. 1, 2, 10 mentioned in the Introduction are to the papers from our group). To make this more clear, we re-phrased the goal statement and explicitly formulated our hypothesis (please see also our answers below and the attached file with the changes highlighted for the specific amendments introduced to the text).
The hypothesis is not clear.
RESPONSE: we re-phrased the goal statement and explicitly formulated our hypothesis (please see also our answers below and the attached file with the changes highlighted for the specific amendments introduced to the text).
Result should be clearly described as per the experimental data obtained.
RESPONSE: we tried to follow the main thread of logic in describing the results, so we introduced changes to the Results section (please see the attached file with the changes highlighted for the specific amendments introduced to the text).
I don’t find any future direction of this work.
RESPONSE: thank you for pointing us to this. We added the outline of future directions to the end of the Discussion section.
Discussion poorly written. Improve it.
RESPONSE: it would be more helpful to have specific indications of places in need of amendment. Still, we have improved the Discussion to the best our abilities to make it more readable.
Overall, this research is quite promising. Although the author did the convincing work, but needs to clarify. Considering all the things, this form of manuscript cannot be published prior to revision. Therefore, I recommended it for major revision.
RESPONSE: we are thankful to the Reviewer for spending time with our manuscript and for its evaluation. We did our best to deal with the issues raised; please see our answers above and the attached file with the changes highlighted for the specific amendments introduced to the text.
Reviewer 2 Report
In this manuscript, the authors investigated the effects of increasing external Pi on cell viability, Pi uptake, and internal P content in the microalga Micractinium simplicissimum. In my opinion, the manuscript is good, very easy to read and the authors presented a good amount of data. I think it is suitable for publication. I found only one or two points in the manuscript that should be corrected. So, I suggest minor revision.
Comments:
Page 4: Please unify the format of the words “starvation” and “deprivation” in the text by using one of the two.
Figure 1b: I cannot see where the i, ii and iii are in the figure.
Page 12: “This fining also corroborates previous” I guess you mean “finding”.
Page 14: “Culture pH was measured with a bench-top pH meter pH410..” I can not find any pH data in the MS. Was the pH in the culture constant during the experiment?
Author Response
In this manuscript, the authors investigated the effects of increasing external Pi on cell viability, Pi uptake, and internal P content in the microalga Micractinium simplicissimum. In my opinion, the manuscript is good, very easy to read and the authors presented a good amount of data. I think it is suitable for publication. I found only one or two points in the manuscript that should be corrected. So, I suggest minor revision.
RESPONSE: We appreciate positive evaluation of our manuscript by the reviewer. Please see the attached file with the changes highlighted for the specific amendments introduced to the text.
Page 4: Please unify the format of the words “starvation” and “deprivation” in the text by using one of the two.
RESPONSE: done.
Figure 1b: I cannot see where the i, ii and iii are in the figure.
RESPONSE: it is indicated in the figure legend as “left to right”: (i) is the leftmost (green), (ii) is in the middle (yellowish), and (iii) is the rightmost (white). To make it more comprehensible, we added the corresponding annotations to the top of Fig. 1b.
Page 12: “This fining also corroborates previous” I guess you mean “finding”.
RESPONSE: yes, indeed. Corrected.
Page 14: “Culture pH was measured with a bench-top pH meter pH410..” I can not find any pH data in the MS. Was the pH in the culture constant during the experiment?
RESPONSE: Yes, it was fairly constant, within 6.7–7.7 range (actually it was specified in the text on page 3: The pH of the cultures remained in the range 6.7–7.7 throughout the experiment, so the Pi added to the cultures is expected to be readily available for the uptake by the microalgal cells.).
Round 2
Reviewer 1 Report
Author did all my concern.